# Tannic Acid-Induced Gelation of Aqueous Suspensions of Cellulose Nanocrystals

**DOI:** 10.3390/polym15204092

**Published:** 2023-10-15

**Authors:** Fengcai Lin, Wenyan Lin, Jingwen Chen, Chenyi Sun, Xiaoxiao Zheng, Yanlian Xu, Beili Lu, Jipeng Chen, Biao Huang

**Affiliations:** 1Fujian Engineering and Research Center of New Chinese Lacquer Materials, College of Materials and Chemical Engineering, Minjiang University, Fuzhou 350108, China; fengcailin@mju.edu.cn (F.L.); linwenyan@stu.mju.edu.cn (W.L.); chenjingwen@stu.mju.edu.cn (J.C.); sunchenyi@stu.mju.edu.cn (C.S.); xxzheng@mju.edu.cn (X.Z.); ylxu@mju.edu.cn (Y.X.); 2College of Material Engineering, Fujian Agriculture and Forestry University, Fuzhou 350108, China; lubl@fafu.edu.cn

**Keywords:** cellulose nanocrystals, tannic acid, hydrogen bonding, hydrogel

## Abstract

Nanocellulose hydrogels are a crucial category of soft biomaterials with versatile applications in tissue engineering, artificial extracellular matrices, and drug-delivery systems. In the present work, a simple and novel method, involving the self-assembly of cellulose nanocrystals (CNCs) induced by tannic acid (TA), was developed to construct a stable hydrogel (SH-CNC/TA) with oriented porous network structures. The gelation process is driven by the H-bonding interaction between the hydroxyl groups of CNCs and the catechol groups of TA, as substantiated by the atoms in molecules topology analysis and FTIR spectra. Interestingly, the assembled hydrogels exhibited a tunable hierarchical porous structure and mechanical moduli by varying the mass ratio of CNCs to TA. Furthermore, these hydrogels also demonstrate rapid self-healing ability due to the dynamic nature of the H-bond. Additionally, the structural stability of the SH-CNC/TA hydrogel could be further enhanced and adjusted by introducing coordination bonding between metal cations and TA. This H-bonding driven self-assembly method may promote the development of smart cellulose hydrogels with unique microstructures and properties for biomedical and other applications.

## 1. Introduction

Self-assembly driven by noncovalent interactions, including hydrogen bonds, ionic bonds, hydrophobic, and van der Waals interactions, provides a powerful strategy to construct functional materials for a variety of applications ranging from biotechnology to photonics [1,2]. Nanoscale materials have been widely used as essential building blocks for self-assembled materials due to their exceptional mechanical, electrical, and optical properties. The intriguing features s of individual nanomaterials can be translated into the macroscopic material properties through self-assembly. Numerous self-assembled nanomaterials have been developed by exploiting gold nanorods [3], graphene [4], carbon dots [5], and silica nanoparticles [6]. In recent years, the macroscopic assemblies of nanoscale biomaterials, such as nanocellulose [7,8], lignin nanoparticles [9,10], and nanochitin [11], have drawn extensive attention due to their abundance, biocompatibility, and capability to form unique structures. Cellulose nanocrystals (CNCs), in particular, are ideal natural building blocks for various self-assembled materials thanks to their abundance of surface hydroxy groups, chiral rod-like nanoscale dimensions, and high aspect ratio [12,13]. A milestone in the development of CNCs self-assembly technology was the observation by Revol et al. that CNC suspensions can form a chiral nematic phase with helical nanorod structure [14]. Thereafter, some reports have been focused on the structural arrangement of CNCs to fabricate novel nanocellulosic materials with attractive optical, mechanical, and heat transport properties [15,16,17,18,19]. However, the overwhelming majority of the previous work on CNC self-assembly materials have been focused on solid films, coatings, and other bulk materials via controlling the microstructure and alignment assembly of the rodlike nanoparticles. In comparison, the feasibility of rod-like CNCs to construct hydrogels with hierarchical porous structures through a self-assembly strategy has been rarely reported before.

Nanocellulose, including microfibrillated cellulose (MFC) and cellulose nanocrystals (CNCs), can be derived from cellulosic biomass through acid hydrolysis, high-pressure homogenization treatment, and enzyme treatment. These nanocellulose materials, sourced from different cellulose materials, exhibit distinct rheological properties due to variations in chemical composition, crystallinity, morphology, and the extracting processes employed [20,21]. Previous research has demonstrated that MFC suspensions display gel-like viscoelastic behavior, even at low concentrations in aqueous or organic media [22,23,24,25]. This behavior arises from the formation of a highly entangled network among MFC particles. The large aspect ratio and high flexibility of MFC particles allow for the easy creation of physical entanglements, even at very low concentrations. Consequently, MFC suspensions exhibit gel-like viscoelastic properties across a range of concentration levels. Conversely, CNC suspensions exhibit elastic gel-like rheological behavior at high concentrations but display the rheological characteristics of a viscous liquid at low concentrations [25,26]. Studies have shown that the rheological behavior of CNC suspensions, as well as the critical concentrations required for hydrogelation, depend on various factors, including the physical or chemical properties of the CNCs, pretreatment process, and post-treatment methods [27,28,29].

CNCs are extracted from abundant cellulosic biomass through an acid hydrolysis process, resulting in the presence of numerous hydroxy groups on their surfaces (Figure 1a) [30,31]. The large number of hydroxyl groups on the surface of CNCs generated a strong H-bonding interaction, which enabled the CNCs to greatly interact with each other as well as the adjacent water molecules. It is worth noting that the needle-like CNCs, with relatively low aspect ratios and high crystallinity, possessed less flexibility compared to MFC. This reduced flexibility limited the formation of physically entangled networks between CNCs [25]. Consequently, in the CNC suspension, the network primarily consisted of hydrogen bonding interactions between CNCs. Therefore, when the CNC concentration was below the critical concentration, the CNC suspension contained a greater number of free water molecules. Additionally, the number of CNC nanoparticles was insufficient to form a continuous sheet-like skeleton structure for the hydrogel. As a result, the CNC suspension exhibited liquid-like rheological properties (Figure 1g). Conversely, when the CNC concentration exceeded the critical concentration (~5.0 wt%), the strong hydrogen bonding interactions between CNC nanoparticles transformed the suspension into an elastic gel-like material with distinct rheological properties (Figure 1g). In parallel, Tannic acid (TA) is a water-soluble natural polyphenol consisting of five pyrogallol and five catechol groups (Figure 1b), providing multiple binding sites with various interactions, such as H-bond, ionic bond, or coordinate bond [32]. Wang et al. have previously explored the construction of self-assembled hydrogels with oriented porous structures using the H-bonding interactions between the catechol/galloyl groups of TA and the hydroxy groups of polyvinyl alcohol (PVA) [33]. Therefore, it is anticipated that TA could serve as an effective gelation binder, inducing the self-assembly of CNCs to produce CNC/TA hydrogels.

In the present work, we report an observation of the H-bonding-induced self-assembly gelation behavior of CNCs and TA. TA serves as a highly effective gelation agent, driven by the formation of multiple hydrogen bonds between TA and CNCs. This interaction leads to the self-assembly of CNC aqueous suspensions into hydrogels that display a well-ordered porous structure. The influence of different CNC contents and CNC:TA mass ratios on the microstructure and storage modulus of these self-assembled hydrogels is also investigated. Furthermore, based on the differential coordination capabilities of various metal ions with TA, the introduction of metal ions allows for the further tuning of the structural stability of the hydrogel. This work provides a theoretical framework for the development of multifunctional cellulose hydrogels based on multiple hydrogen bonds.

## 2. Materials and Methods

### 2.1. Materials

Bleached bamboo pulp (α-cellulose content > 95%) was provided by Nanping Paper Co., Ltd. (Nanping, Fujian, China). Tannic acid, aluminum nitrate nonahydrate, cupric nitrate trihydrate, iron (III) nitrate nonahydrate, silver nitrate, and zinc nitrate hexahydrate were purchased from Aladdin Industrial Corporation (Shanghai, China). The distilled water was used for the preparation of aqueous solutions.

### 2.2. Synthesis of SH-CNC/TA Hydrogels

Cellulose nanocrystal (CNC) suspension with the concentration of 7.5 wt% was first prepared from bamboo cellulose pulp according to our previous study [34]. A certain amount of CNC suspension and tannic acid (TA) was first dispersed in an appropriate amount of deionized water with total mass of 5.0 g in a glass container, then followed by ultrasonication to form a homogeneous dispersion. Hydrogelation occurred gradually upon the addition of tannic acid. Subsequently, the obtained mixture was stored at room temperature for 12 h without stirring for further self-assemble into a hydrogel. The resulting hydrogel with different contents of CNCs and TA were labeled as SH-αCNCx/TAy, where α is the weight content (wt%) of the CNCs, and x:y is the mass ratio of CNC: TA. For example, SH-4CNC1/TA3 means that the concentration of CNC in the hydrogel is 4 wt%, and the mass ratio of CNC:TA is 1:3. For comparison, pure CNC hydrogels were also prepared and denoted as SH-CNC.

### 2.3. Synthesis of SH-CNC/TA-M^n+^ Hydrogels

The SH-CNC/TA-M^n+^ hydrogels were assembled by dispersing CNC suspension, TA, and metal ion (M^n+^) in deionized water with total mass of 5.0 g in a glass container followed by ultrasonication to form a homogeneous dispersion. The CNC concentration was 4.0 wt%, the mass ratio of CNC:TA was 1:1, and the molar ratio of TA to metal ion was fixed at 1:1.5. Then the obtained mixture was stored at room temperature for 12 h for further self-assembly into a hydrogel. The resulting hydrogel with different metal ions was labeled as SH-4CNC1/TA1-M^n+^, where M^n+^ is the metal ions (including Al^3+^, Fe^3+^, Zn^2+^, Cu^2+^, Ag^1+^).

### 2.4. Characterizations

The microstructure of the SH-CNC/TA hydrogels was observed by scanning electron microscope (SEM, SU8010, Hitachi, Japan) at an acceleration voltage of 40 kV. Before observation, hydrogel samples were freeze-dried and coated with gold. The morphology of CNCs was carried out on a Hitachi H-7650 (Japan) electron microscope at an acceleration voltage of 80 kV. FTIR spectra of the samples in KBr discs were performed on a NICOLET 380 FTIR spectrometer (Thermo Electron Instruments Co., Ltd., Waltham, MA, USA) in the range of 400–4000 cm^−1^ with 32 scans.

### 2.5. Rheological Measurement

Rheological behaviors of SH-CNC/TA hydrogels were analyzed using a rheometer Rotational Rheometer MARS III Haake (Thermo Scientific, Bremen, Germany) equipped with a parallel plate geometry (35 mm in diameter) and a 1 mm gap. Oscillatory frequency sweeps (strain = 1%) was conducted from 0.01 to 10 Hz at 25 °C. The complex modulus (*G**) and complex viscosity (*η**) were calculated by Equations (1) and (2).
(1)$$G^{*}=\sqrt{G^{\prime 2}+G^{\prime \prime 2}}$$
(2)$$\eta^{*}=G^{*}/f$$
where *G*′ is the storage modulus, *G*″ is the loss modulus, and f is the angular frequency.

Recovery rests were carried out by straining sample to failure with increasing sweep strain from 0.1–1000%, and then using the time sweep module to record the recovery of the storage modulus (*G*′) and loss modulus (*G’’*) as a function of time. Moreover, the *G*′ and *G*″ dependence of time in the continuous step strain for SH-CNC/TA hydrogels were carried out with the change of amplitude oscillatory force from strain of 1% to 300% under the same frequency of 1.0 Hz.

### 2.6. Model Development

Molecular simulation calculations were employed to quantitatively analyze the H-bonding interactions in SH-CNC/TA hydrogels. All theoretical calculations were conducted using Gaussian 16 A. 03 software, employing the M06-2X hybrid functional and the 6-311*G** basis set. A DFT-D3 dispersion correction was applied to improve the accuracy of describing weak interactions. AIM (Atoms in Molecules) analysis and IGM (Independent Gradient Model) analysis were performed using Multiwfn 3.7 software, followed by the visualization of intermolecular interactions using VMD (Visual Merchandise Design).

## 3. Results and Discussion

### 3.1. SH-CNC/TA Hydrogels Assembly and Proposed Formation Mechanism

In the present work, cellulose nanocrystals were prepared by the hydrolysis of bleached bamboo pulp (α-cellulose content > 95%, the weight average molecular weight and the number average molecular weight were 203,000 and 58,000, respectively) using 50% w/w sulfuric acid at 65 °C with ultrasonic treatment for 120 min according to our previous work [34]. As shown in Figure 1f, the cellulose nanocrystals (CNCs) prepared for this study display a rod-like structure with lengths ranging from 200–500 nm and diameters of 25–50 nm. These CNCs consist of β (1–4) linked D-glucose units and offer tunable surfaces chemistries due to their abundant hydroxyl groups (Figure 1a). Figure 1b illustrates the dendritic structure of the TA molecule, characterized by a five-polyphenol-arm structure with multiple catechol groups. These catechol groups enable TA to readily form bonds with other materials through various interactions, including hydrogen and ionic bonding. As shown in Figure 1c–e, when mixing the suspensions of CNCs (4.0 wt%) with TA (CNC:TA = 1:2), the self-assembly of an SH-CNC/TA hydrogel with highly porous 3D network architecture (Figure 1e) gradually initiates. This initiation is attributed to the strong H-bonding interactions between hydroxyl and catechol groups (Figure 1d). As a result, TA significantly increases the viscosity of CNC suspensions, making it a valuable candidate for use as an effective gelation binder in the self-assembly of CNCs.

To further understand the gelation process, the influence of the concentration of CNCs, and the mass ratio of CNCs to TA on the formation of SH-CNC/TA hydrogels were investigated. Figure 1l presents the rheological diagram of CNCs and TA in the hydrogel formation. This result shows that increasing mixture concentration enhances the interaction between CNCs and TA to make the mixture system achieve hydrogelation more easily, and the critical CNC concentration (C*) for hydrogelation is ~2.0 wt%. Below the C*, the number of CNC nanoparticles is insufficient to contact to form the continuous sheet-like skeleton structure of hydrogel even under a high concentration of TA. Figure 1h–k presents the TA-induced gelation of CNC suspensions with different concentrations (from 1.0 to 4.0 wt%). When the concentration of CNCs is 1.0 wt%, no gelation was observed under experimental condition even though the mass ratio of CNC:TA is up to 1:10 (Figure 1h). As CNC concentration increased from 2.0 wt% to 4.0 wt% (Figure 1i–k), the mass ratio of CNCs to TA for initiating gelation was 1:10, 1:3, and 1:0.7, respectively. These results clearly indicate that the gelation of CNC suspensions is a concentration-dependent process. At low concentration, TA molecules are only attached to the surface of an individual CNC nanoparticle rather than binding them together, leading to a low cross-link density between the CNC nanoparticles. This low H-bond density within the system is insufficient to support the gelation of CNC suspensions. Consequently, increasing the H-bonding interactions between CNC nanoparticles is essential to achieve the necessary crosslink density for the assembly of SH-CNC/TA hydrogels.

The H-bonding interactions between TA and CNCs were confirmed by FTIR spectra (Figure 2). It is well-known that the formation of H-bonding interactions reduced the force constants of the -OH groups, resulting in a redshift of -OH vibrational frequencies [22]. Stronger H-bonding interactions result in a more pronounced shift in vibrational frequency [35]. In the FTIR spectra of CNCs, the strong adsorption bands at 3432 cm^−1^ and 1641 cm^−1^ are attributed to the hydroxy groups (O-H) stretching vibration and bending vibration, respectively [36,37,38]. After introducing TA, the O-H stretching bands of the SH-CNC/TA hydrogels at the mass ratios (CNC:TA) of 1:0.7, 1:1, and 1:3 are shifted to a lower wavenumber of 3409 cm^−1^, 3398 cm^−1^, and 3388 cm^−1^, respectively. In addition, the O-H bending vibration picks of SH-CNC/TA hydrogels presented a similar redshift. The absorption peak of O-H for SH-CNC/TA hydrogels shifts from 1630 cm^−1^ to 1616 cm^−1^ with the mass ratio of CNC:TA increasing from 1:0.7 to 1:3. The significant redshift of the vibration peaks indicated the formation of strong H-bonding interactions between CNCs and TA, and, moreover, the intensity of H-bonding interactions of SH-CNC/TA hydrogels could be tunable by adjusting the mass ratio of CNC:TA. Furthermore, the characteristic vibration peaks of TA appeared at 1715 cm^−1^ (C=O), 1537 cm^−1^, 1448 cm^−1^ (aromatic C-C), and 762 cm^−1^ (phenolic hydroxyl) [39,40], further indicating the successful assembly of the SH-CNC/TA hydrogel.

Theory calculation is used to further uncover the hydrogelation mechanism. Firstly, the non-covalent interactions of CNC-TA, CNC-CNC, CNC-H_2_O, and TA-H_2_O complexes within SH-CNC/TA hydrogels in real space were directly visualized by employing Visual Merchandise Design (VMD) software (v 1.9.3). As depicted in Figure 3, a continuous isosurface with a bright blue color appears between oxygen and hydrogen atoms of CNC, TA, and H_2_O molecules, which corresponds to the attractive H-bonding interactions, indicating that the SH-CNC/TA hydrogel is assembled mainly by H-bonding interaction. Then, the number and interaction strength of H-bond between CNC, TA, and H_2_O molecules were further quantified by the atoms in molecules (AIM) methodology. According to AIM theory, the ρ value is used to describe the strength of the H-bond; a stronger H-bond is associated with a larger ρ value [41]. The optimized configurations of CNC-TA, CNC-CNC, CNC-H_2_O, and TA-H_2_O complexes are presented in Figure 4, and their corresponding AIM topology calculations are listed in Table 1. It is shown that the maximum ρ value of 0.0332 a.u. for the CNC-TA H-bond is larger than that of CNC-CNC (0.0281 a.u.), CNC-H_2_O (0.0225 a.u.) and TA-H_2_O (0.0232 a.u.) H-bonds and the H-bond number of the CNC-TA complex are more than that of other complexes, suggesting that the intensity of H-bond between CNCs and TA is the strongest in the SH-CNC/TA hydrogel. Moreover, the absolute value of the interaction energies of complexes within the hydrogel is in the order of CNC-TA (129.891 kJ/mol), CNC-CNC (67.568 kJ/mol), TA-H_2_O (32.667 kJ/mol) and CNC-H_2_O (16.706 kJ/mol), which manifests that the CNC will be more favorable to combine with TA molecule. In summary, the assembly of SH-CNC/TA hydrogels was primarily driven by hydrogen bonding interactions between CNC and TA molecules. Upon adding TA to the CNC suspensions, TA molecules first interacted with CNC nanoparticles and subsequently bound these CNC nanoparticles together through multiple hydrogen bonding interactions. This process ultimately led to the gelation of the CNC suspensions.

### 3.2. Morphology and Microstructure of SH-CNC/TA Hydrogel

It is well known that the concentration of components and their interactions are critical factors for controlling the microstructure of a hydrogel. To investigate the effect of TA on the assembling behavior of CNC nanoparticles, pure CNC suspensions, and SH-CNC/TA, the hydrogels were freeze-dried. As shown in Figure 5a–d, the pure CNC suspensions in all concentrations (ranging from 1.0 wt% to 4.0 wt%) exhibited a sheet-like structure with a disorderly and scattered distribution. This can be ascribed to the low viscosity of CNC suspensions and relatively weak interactions between nanoparticles, resulting in a lack of interconnection among nanocellulose sheets and the propensity for the 3D structure to collapse during freeze-drying. In contrast, after self-assembling triggered by TA (CNC:TA = 1:1), the SH-CNC/TA hydrogels displayed a lamellar structure with thin membrane layers and good orientation at the low CNC concentration (1.0 wt% and 2.0 wt%, Figure 5e,f). Subsequently, they gradually transitioned into a regularly oriented 3D macroporous structure as the CNC concentration increased from 3.0 wt% to 4.0 wt% (Figure 5g,h). This transformation is likely a result of the robust multiple H-bonding interactions between TA and CNC particles, facilitating their assembly into a stable and oriented 3D porous structure that remains intact during freeze-drying.

Figure 6 presents the influence of the mass ratio of CNC:TA on the morphology of SH-CNC/TA hydrogels (CNC concentration fixed at 4.0 wt%). As the mass ratio of CNC:TA increased from 1:0.7 to 1:3, the 3D macroporous structure in SH-CNC/TA hydrogel became more uniform, compact, and exhibited a narrow size distribution, with the pore walls aligning in a direction parallel to one another. This unique structure formation process was driven by the H-bonding between the surfaces of the CNC nanoparticles. At high mass ratio levels, more TA was introduced onto the surfaces of the CNCs, resulting in stronger interactions between CNC nanoparticles and thus a more compact porous architecture. The distinctive morphology differences observed above indicate that TA induced multiple H-bonding interactions play a critical role in the self-assembly properties of SH-CNC/TA hydrogels, with significantly stronger H-bonding interactions forming between CNC particles at high CNC:TA mass ratio.

### 3.3. Dynamic Rheological Properties of SH-CNC/TA Hydrogels

The intensity of H-bonds, which depends on the mass ratio of CNC:TA, not only determines the hydrogelation and morphology of SH-CNC/TA hydrogels, but also affects its mechanical properties. As we know, hydrogen bonding interactions are relatively weaker compared to covalent bonds. As a result, the SH-CNC/TA hydrogels formed in this study exhibited lower mechanical strength, making conventional mechanical performance tests challenging to perform. To address this limitation, we employed dynamic rheological analysis as a complementary method to investigate the structural stability of SH-CNC/TA hydrogels. The hydrogels assembled at the mass ratios (CNC:TA) of 1:0.7, 1:1, 1:2, and 1:3 with a CNC concentration of 4.0 wt% were chosen as the model hydrogel for dynamic rheological measurement. As depicted in Figure 7a, in the case of CNC suspensions (4.0%), both storage modulus (*G′*) and loss modulus (*G″*) exhibited noticeable frequency dependence, with G′ being smaller than G″ from 0.1 to 3.0 Hz, indicating a lack of crosslink-like interactions between CNC nanoparticles [41]. In contrast, *G′* > *G″* was observed for all the SH-CNC/TA hydrogels, and both moduli showed minimal dependence on frequency over the entire frequency range, suggesting the stability of the hydrogel network. Furthermore, both the *G′* and *G″* values increased with the increase in TA content. For the SH-4CNC1/TA3 hydrogel, in particular, maximum values of 8.1 Pa (*G′*) and 3.3 Pa (*G″*) were achieved, respectively. Because there is no chemical crosslinking between CNCs and TA, this higher modulus was probably due to the formation of a more compact 3D network structure of the SH-CNC/TA hydrogels, facilitated by the introduction of higher TA content, which readily couple with CNC nanoparticles via intensive H-bonding interactions. Furthermore, as shown in Figure 7b,c, the complex modulus (*G**) and complex viscosity (*η**) of SH-CNC/TA hydrogels also reached their highest values when the mass ratio of CNC:TA increased to 1:3, further indicating the critical role of TA in enhancing the viscoelasticity and H-bonding assembly of these hydrogels.

It is well known that every nanocellulose gel is capable of self-healing. In the present study, our focus was specifically on investigating the self-healing properties of hydrogels formed through multiple H-bonding interactions between TA and CNCs. These H-bonds between CNCs and TA are dynamically reversible and endow self-healing abilities to the SH-CNC/TA hydrogels, implying that they spontaneously repair after damage. As shown in Figure 7d, the SH-4CNC1/TA3 hydrogel was cut into two halves with a spatula, and then the separated pieces were put back into contact, allowing the reformation of the broken H-bonds at room temperature. After curing for 2 min, the damaged interface automatically self-healed. The self-healing properties were further investigated using rheology measurements. The hydrogel (SH-4CNC1/TA3) was first subjected to increasing strains until fracture, followed by monitoring the recovery of mechanical properties (Figure 7e). Upon the application of a large shearing strain, both the *G′* and *G″* values reduced significantly and *G′* became lower than *G″*, suggesting the disruption of hydrogel network, which converted into a sol state. Once the strain returned to 1%, both *G′* and *G″* fully recovered to their initial values instantaneously, showing the rapid self-recovery ability of SH-CNC/TA hydrogels. Meanwhile, repeated alternate step strain tests (strain = 1% or 300%) were also applied to the SH-4CNC1/TA3 hydrogel. As shown in Figure 7f, when subjected to a large strain of 300%, G′ immediately decreased from 8.4 Pa to around 1.3 Pa, along with an inversion of *G″* exceeding *G′*. Once the strain returned to 1%, the hydrogel instantly recovered to their original modulus, regardless of the loading period, further indicating the quick reconstruction of reversible H-bond networks.

### 3.4. Effect of Metal Cations on the Rheological Properties of SH-CNC/TA Hydrogels

Interestingly, the structural stability of the CNC hydrogel could be further enhanced and adjusted by introducing metal ions. As shown in Figure 8, we introduced metal ions (with a TA to metal ion molar ratio fixed at 1:1.5) into the SH-4CNC1/TA1 self-assembly system to construct a secondary noncovalent network based on metal–catechol coordinate bonds, resulting in stable SH-CNC/TA-M^n+^ hydrogels. As depicted in Figure 8a, TA molecules first interacted with CNC nanoparticles via hydrogen bonds, and then TA bound CNC particles together with the assistance of metal ions via coordination bonds. Figure 8b presents the resulting SH-CNC/TA-M^n+^ hydrogels with different metal ions, and the change hydrogel color indicates the formation of metal-catechol coordinate bonds. The influence of metal cations on the *G′* and *G″* of SH-CNC/TA-M^n+^ hydrogels is depicted in Figure 8c,d. As expected, the G′ values of SH-CNC/TA-M^n+^ hydrogels are higher than that of *G″*, and both modules remain almost unchanged over the entire frequency range, indicating the existence of a stable hydrogel network in the hydrogels. The *G′* values of the SH-CNC/TA-M^n+^ hydrogels exhibit as being 21 (62.8 Pa for SH-4CNC1/TA1-Ag^1+^) to 125 (367.7 Pa for SH-4CNC1/TA1-Al^3+^) times higher than that of the SH-4CNC1/TA1 (2.93 Pa) hydrogel. This enhancement can be attributed to the synergistic interactions between the hydrogen bonds and coordination bonds. Moreover, the obtained SH-CNC/TA-M^n+^ hydrogels span a wide range of storage modulus from 367.7 to 62.8 Pa, in the order of Al^3+^ > Fe^3+^ > Zn^2+^ > Cu^2+^ > Ag^1+^, providing a method for assembling CNC hydrogels with tunable mechanical properties. Due to the absence of covalently derived cross-links, the significant improvements in mechanical moduli are likely attributed to the different coordination ability between TA and metal ions [42].

## 4. Conclusions

In summary, we have demonstrated the gelation behavior of CNC aqueous suspensions induced by TA. It was observed that, as the CNC concentration increased from 2.0 wt% to 4.0 wt%, the required mass ratio of CNCs to TA for initiating gelation transitioned from 1:10, to 1:3, and ultimately to 1:0.7. The driving force behind gelation is proposed to be the H-bonding interaction between hydroxyl groups of CNCs and catechol groups of TA, a proposition substantiated by AIM topology analysis and FTIR spectra. Interestingly, these multiple H-bonding interactions impart the SH-CNC/TA hydrogel with uniform, compact, and oriented porous network structures, which can be easily adjusted by varying the mass ratio of CNCs to TA. In contrast to CNC suspensions (4.0%), the obtained SH-CNC/TA hydrogels exhibited stable gel-like viscoelastic properties with minimal frequency dependence and increased moduli as tannic acid content rose, reaching maximum values of 8.1 Pa for *G′* and 3.3 Pa for *G″* in the SH-4CNC1/TA3 hydrogel. Due to the dynamic nature of the H-bonded network, these SH-CNC/TA hydrogels also exhibited rapid self-healing capabilities. Furthermore, the coordination bonding between metal cations and TA can further enhance the storage modulus of the hydrogel, with SH-CNC/TA-M^n+^ hydrogels showing varying storage moduli (ranging from 367.7 to 62.8 Pa) in the order of Al^3+^ > Fe^3+^ > Zn^2+^ > Cu^2+^ > Ag^1+^. We anticipate that H-bonding, although weaker than covalent bonds, may play a critical role in providing the dynamic and reversible interactions necessary for energy dissipation and self-recovery in advanced nanocomposites where CNC is applied. While the SH-CNC/TA hydrogel demonstrates interesting performances, there is a need for further enhancement in its mechanical performance and multifunctionality. Our future research will focus on augmenting the hydrogel’s mechanical characteristics through the introduction of additional components and modifying the assembly conditions, thus enabling multifunctional modifications.

## Figures and Tables

**Figure 1 polymers-15-04092-f001:**
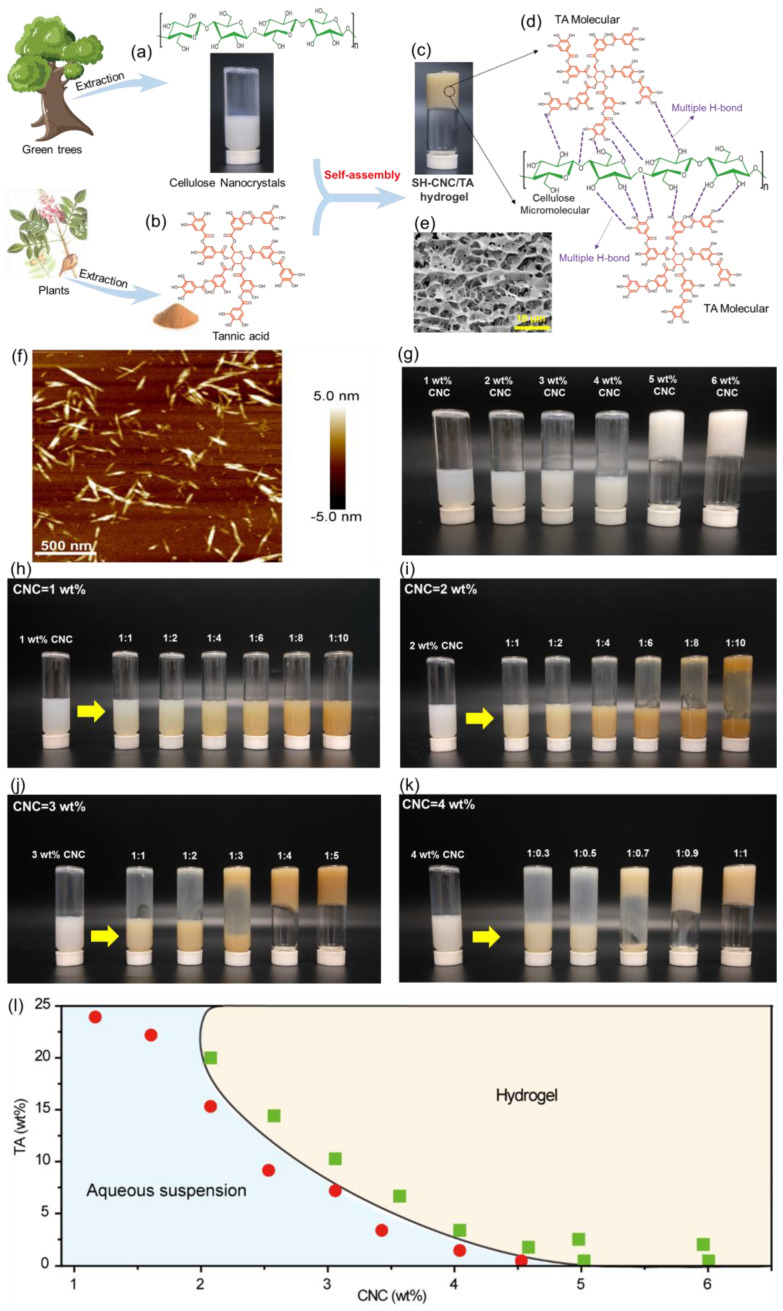
Schematic for the formation of SH-CNC/TA hydrogels. Chemical structures and morphologies of CNCs (**a**) and TA (**b**). (**c**,**d**) Schematic representation of the proposed gelation mechanism between CNCs and TA (CNC:TA = 1:2). (**e**) SEM image of SH-CNC/TA hydrogel presents a highly porous architecture. (**f**) AFM image of the prepared CNCs. (**g**) Photographs of CNC suspensions with the concentrations from 1.0 wt% to 6.0 wt%. (**h**–**k**) Photographs of TA-induced self-assembly hydrogelation of CNC suspensions with different CNC concentrations and mass ratio of CNCs to TA. (**l**) Rheological diagram of CNC/TA mixture.

**Figure 2 polymers-15-04092-f002:**
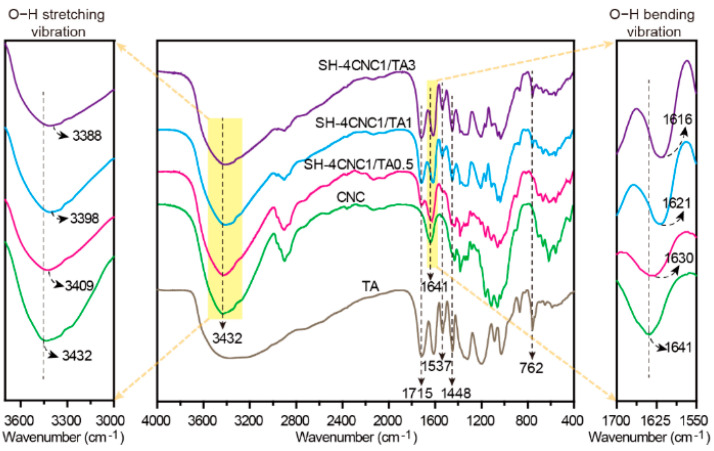
FTIR spectra of CNCs, TA, and SH-CNC/TA hydrogels.

**Figure 3 polymers-15-04092-f003:**
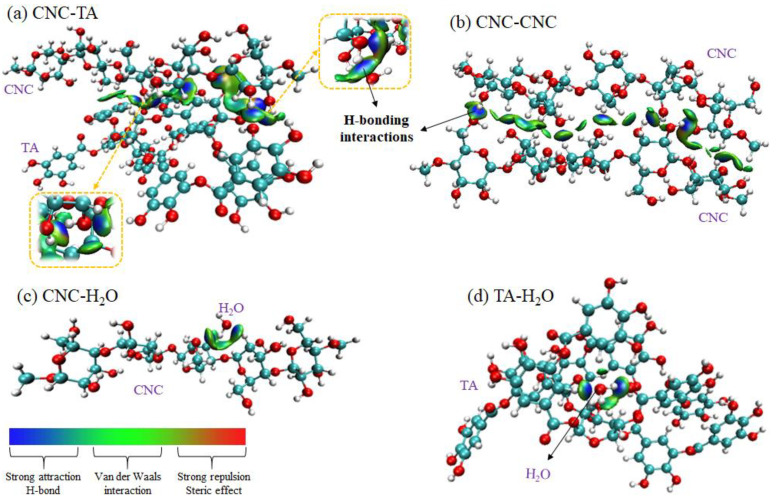
3D intermolecular non-covalent interaction plots of CNC-TA (**a**), CNC-CNC (**b**), CNC-H_2_O (**c**), and TA-H_2_O (**d**).

**Figure 4 polymers-15-04092-f004:**
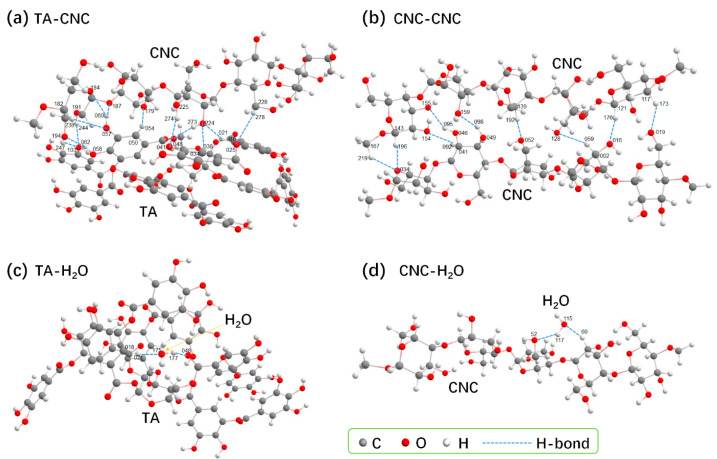
The optimized configurations for CNC-TA (**a**), CNC-CNC (**b**), CNC-H2O (**c**) and TA-H_2_O (**d**). H-bonds are indicated by dotted lines. Each TA molecule forms 12 H-bond interactions with CNC, and only 9 H-bonds between 2 CNC molecules. Each TA or CNC molecule forms 2 H-bonds with H_2_O molecule, respectively.

**Figure 5 polymers-15-04092-f005:**
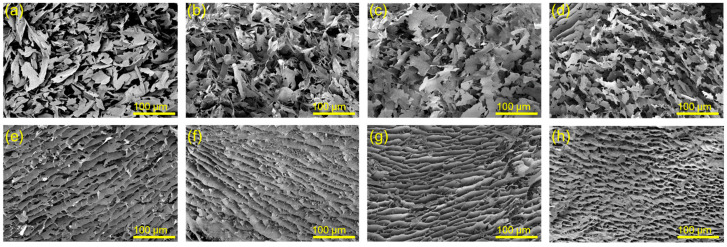
SEM images of freeze-dried CNC suspensions at the concentration of 1.0 wt% (**a**), 2.0 wt% (**b**), 3.0 wt% (**c**), 4.0 wt% (**d**), and freeze-dried TA-induced self-assembled CNC suspensions (CNC:TA = 1:1) at the concentration of 1.0 wt% (SH-1CNC1/TA1) (**e**), 2.0 wt% (SH-2CNC1/TA1) (**f**), 3.0 wt% (SH-3CNC1/TA1) (**g**), 4.0 wt% (SH-4CNC1/TA1) (**h**).

**Figure 6 polymers-15-04092-f006:**
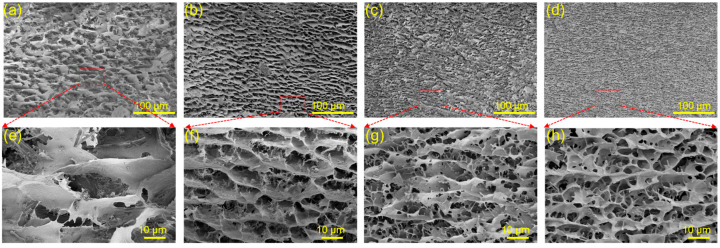
SEM images of SH-CNC/TA hydrogel: (**a**,**e**) SH-4CNC1/TA0.7, (**b**,**f**) SH-4CNC1/TA1, (**c**,**g**) SH-4CNC1/TA2, (**d**,**h**) SH-4CNC1/TA3.

**Figure 7 polymers-15-04092-f007:**
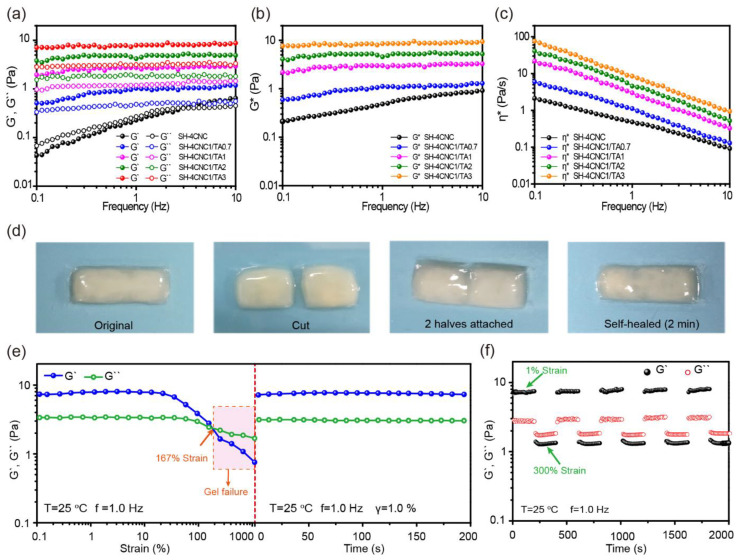
(**a**) Storage modulus (*G′*) and loss modulus (*G″*) of SH-CNC/TA hydrogels versus frequency (25 °C, strain = 1%). Complex modulus (*G**) (**b**) and complex viscosity (*η**) (**c**) of SH-CNC/TA hydrogels versus frequency (25 °C, strain = 1%). (**d**) Photographs were showing the self-healing ability of SH-4CNC1/TA3 at ambient temperature. (**e**) G′ and G″ of SH-4CNC1/TA3 hydrogel versus strain (frequency = 1.0 Hz) (**left**) and immediate recovery from the 1000% strain deformation (**right**). (**f**) *G′* and *G″* recorded during the cyclic strain changes between a small oscillation force (strain = 1%, frequency = 1.0 Hz) and a large one (strain = 300%, frequency = 1.0 Hz).

**Figure 8 polymers-15-04092-f008:**
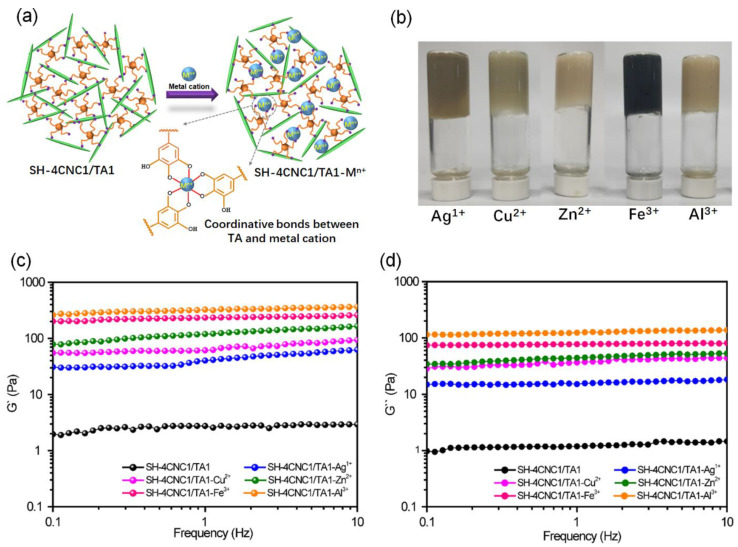
(**a**) Schematic illustration of the formation mechanism of SH-CNC/TA-M^n+^ hydrogels. (**b**) Photos of SH-CNC/TA-M^n+^ hydrogels with different metal ions. Storage modulus (G′) (**c**) and loss modulus (G″) (**d**) of SH-CNC/TA-M^n+^ hydrogels versus frequency (25 °C, strain = 1%).

**Table 1 polymers-15-04092-t001:** The electron density (ρ) at the bond critical point and interaction energies (Eint) for the intermolecular interaction of CNC-TA, CNC-CNC, CNC-H_2_O, and TA-H_2_O complex.

Complex	Bond	ρ (a.u.)	E_int_ (kJ/mol)
CNC-TA	C_228_-H_278_…O_025_ C_018_-H_021_…O_224_ C_034_-H_038_…O_224_ O_224_-H_273_…O_048_ O_225_-H_274_…O_041_ C_050_-H_054_…O_179_ O_057_-H_060_…O_187_ O_057_-H_060_…O_184_ C_182_-H_238_…O_057_ O_058_-H_062_…O_194_ O_194_-H_247_…O_103_ O_191_-H_244_…O_103_	0.0097 0.0197 0.0151 0.0297 0.0083 0.0227 0.0139 0.0141 0.0068 0.0332 0.0209 0.0208	−129.891
CNC-CNC	C_167_-H_219_…O_034_ C_143_-H_196_…O_034_ C_041_-H_092_…O_154_ O_046_-H_095_…O_155_ O_049_-H_098_…O_159_ C_139_-H_192_…O_052_ C_002_-H_059_…O_128_ C_121_-H_176_…O_016_ C_117_-H_173_…O_019_	0.0151 0.0173 0.0281 0.0132 0.0267 0.0207 0.0194 0.0218 0.0169	−67.568
TA-H_2_O	C_018_-H_021_…O_175_ O_175_-H_177_…O_048_	0.02325 0.02312	−32.667
CNC-H_2_O	C_003_-H_060_…O_115_ O_115_-H_117_…O_052_	0.0197 0.0225	−16.706

## Data Availability

The data presented in this study are available on request from the corresponding author.

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
