# Peer review of "Tannic Acid-Induced Gelation of Aqueous Suspensions of Cellulose Nanocrystals"

_polymers, 2023, doi:10.3390/polym15204092_

Round 1

Reviewer 1 Report

This is an interesting study about tannic acid-induced gelation of aqueous suspensions of cellulose nanocrystals. I strongly suggest it for publication in Polymers after the following minor points are addressed.

1. Line 37-39, one review (Current opinion in biotechnology 39, 76-88) related to this point should be included to support such a claim.

2. What does SH-CNC represent?

3.  Formatting issues. For example, line 98, 'Mn+' should be 'Mn+'.

4. The metal ions involve the construction of the hydrogel. The authors should mention this in the abstract.

Minor editing of English language required

Reviewer 2 Report

The article by Lin F. et al. describes the formation of strong gels when tannic acid is added to an aqueous dispersion of cellulose nanocrystals. The authors obtain dispersions with different ratios of cellulose and tannic acid and then measure IR spectra, perform computer modeling, measure the storage and loss moduli of the gels, and study their morphology after freeze-drying. Additionally, the authors investigate the addition of different metal cations to the stiffness of the gels. As a result, the authors show that the addition of tannic acid promotes gelation, and the addition of multivalent metal cations further enhances it.

Overall, the article is interesting, although there is a serious lack of flow curves for the systems under study. The authors consider gels but do not give any flow curves. Gels have a yield stress. The authors only show this visually by inverting the jars with the samples, which is not scientific, although illustrative. Viscoelasticity alone is insufficient, while yield stress measurements could say more about the strength of the gels.

In addition, there are the following comments.

Line 50: “Because of the hydrogen bonding (H-bonding) interactions between the surface hydroxy groups of CNC, spontaneous gelation will occur at high particle concentrations (~5.0 wt%)”. 5% is a very high concentration for nanocellulose gelation. Nanocellulose forms a gel at much lower concentrations, such as 0.3% in aqueous media (10.1021/acs.energyfuels.0c02797) and 1% in organic media (10.1016/j.triboint.2022.108080). Why do these nanocellulose crystals not do this? The authors should comment on this.

Line 111: “35 mm in diameter”. The gap between the plates needs to be specified.

Line 111: “strain=10%”. This is a very large strain amplitude that most likely partially destroyed the gel structure. Usually, the linear strain for nanocellulose gel is 0.1%, maybe 1% at most. Moreover, in Figure 7 the authors write that the strain was 1%.

Lines 115-116: f is the angular frequency, or the formula (2) is written with an error.

Line 143: “Consequently, TA significantly increases the viscosity of CNC suspensions, ultimately leading to the hydrogelation of CNC.” This is may be an incorrect interpretation. TA most likely increases the yield stress of the nanocellulose gel. Nanocellulose at a concentration of 4% may form a gel even without TA, but it is a very weak gel with a low yield stress. As a result, when the jar with this gel is turned upside down, its structure collapses under gravity because the yield stress is low. The addition of TA increases the interaction between the nanocellulose particles, and the yield stress increases, resulting in a strong gel that does not collapse when inverted. Most likely, TA is a gelation enhancer, rather than a gelator.

Line 148: “Chemical structures and morphologies of CNC (a)”. The authors provide an AFM image for cellulose crystals but do not write about their dimensions. The dimensions (length and diameter) of nanocrystals should be described in the article.

Lines 152, 157: “(g) Phase diagram of CNC/TA mixture. (h-k)”, “the phase diagram”. It is not a phase diagram because there are two phases (water and cellulose) in both cases (hydrogel and liquid). There is no change in phase state upon gelation. This diagram can be called a rheological diagram or a nominal state diagram, but not a phase diagram.

Line 300: “endow self-healing abilities”. Every nanocellulose gel is capable of self-healing. There is nothing unique in this. It is a trivial fact.

Line 357: “particularly in the biomedical fields”. How specific? Without specifics, this statement is meaningless.

The English language requires moderate editing.

Round 2

Reviewer 2 Report

The authors made corrections to the manuscript but did not complete the work.

First of all, it is puzzling that the authors gave good responses, but did not provide them in the manuscript. If a reviewer has these questions, then any reader may have them. Therefore, it is necessary not only to respond to the reviewer, but it is essential to make these responses in the form of corrections in the manuscript. For example, the authors write in the responses about the difference between CNC and MFC (“Nanocellulose, including microfibrillated cellulose (MFC) and cellulose nanocrystals (CNC), can be extracted…”) but do not write about it in the article. This response text should be provided in the introduction. This is particularly important since the introduction to this article is relatively small and contains only 21 references, which is remarkably few on this hot topic. The introduction needs to be expanded, and the authors' response is suitable for this. The same is true for other responses about dynamic rheological analysis and self-healing. Responses should not only be made to the reviewer but all responses should be transferred to the article.

Further, the conclusions of the article should also be improved. The current conclusions do not contain any concrete details and are merely of a narrative nature. The conclusions should include clear findings. In addition, the authors should write about the limitations and shortcomings of the current work and plans for future studies.

In addition, on Line 134: “As shown in Figure 1a, the cellulose nanocrystals (CNC) prepared for this study display a rod-like structure with lengths ranging from 200-500 nm and diameters of 25-50 nm.” In Figure 1a, there are no dimensional scales, and the figure itself is very small to judge the size of the crystals. The figure should be corrected or even given as a separate figure.

Thus, the article still needs to be improved in terms of expanding the introduction and conclusions.

The English language requires moderate editing.

Round 3

Reviewer 2 Report

The authors have made the necessary corrections to the manuscript for its publication.

The English language requires moderate editing.